# Values and Risk Perception Shape Canadian Dairy Farmers’ Attitudes toward Prudent Use of Antimicrobials

**DOI:** 10.3390/antibiotics11050550

**Published:** 2022-04-20

**Authors:** Claudia Cobo-Angel, Basem Gohar, Stephen J. LeBlanc

**Affiliations:** 1Department of Population Medicine, University of Guelph, Guelph, ON N1G 2W1, Canada; bgohar@uoguelph.ca (B.G.); sleblanc@ovc.uoguelph.ca (S.J.L.); 2CLEV Research Group, Universidad de Caldas, Manizales 170004, Colombia

**Keywords:** antimicrobial stewardship, Twenty Item Values Inventory, antibiotic, One Health, antimicrobial resistance, antimicrobial use

## Abstract

Antimicrobial resistance (AMR) is an important challenge in public health. Ensuring rational antimicrobial use (AMU) on farms is one of the key components of antimicrobial stewardship. We aimed to describe a sample of Canadian dairy farmers’ personal factors for AMUand their AMR risk perception, and to associate these factors with their attitude toward promotion of prudent AMU. We distributed an online survey among dairy farmers in Ontario and Atlantic Canada (New Brunswick, Nova Scotia, and Prince Edward Island). The questionnaire was designed to solicit information on dairy farmers’ AMU decision-making process, attitudes toward AMU reduction, awareness of AMR, and individual values. We performed a factor analysis on 15 statements related to AMR awareness and AMU reduction and used a logistic regression model to identify variables associated with the probability of disagreeing with the need to increase promotion of responsible AMU in the dairy industry. Respondents’ (n = 193) previous experience was the main reason to select an antimicrobial treatment for their cattle. We identified four groups of factors related to knowledge, risk perception, and emotional states among respondents. To the question “Should there be more initiatives to promote responsible use of antibiotics in the dairy industry?” 23% of respondents answered no, which was associated in a logistic regression model with being a farm owner, having a tie-stall barn, and considering their own experience as the most important factor in selecting antimicrobial treatments. The score for the conservation value dimension and score for a factor described as sense of responsibility when using antimicrobials were also retained in the final model. Our results indicate that tailored strategies to promote prudent AMU on dairy farms may be preferable to a generic strategy because there are individual differences in attitudes, values, and AMR awareness that shape AMR risk perception and willingness to modify current AMU practices.

## 1. Introduction

Antimicrobial resistance (AMR) is a substantial public health challenge [1]. Infections with resistant pathogens result in an increased risk of disease spread, severe illness, and death, raising the social and economic costs of disease [2]. The overuse and misuse of antimicrobials have accelerated the development of resistant microorganisms [3]. However, AMR inevitably develops over time as a consequence of antimicrobial use (AMU) in humans and animals, even when used prudently, because exposure to antimicrobials increases the selection pressure for resistance mechanisms and gene transfer between bacteria [2]. Available literature suggests a positive correlation between the amount of antimicrobials used in food-producing animals and the prevalence of AMR in veterinary pathogens [2], whereas the link between AMU in animals and AMR in human pathogens has not been quantified [4]. Nevertheless, there is global pressure to reduce AMU and improve antimicrobial stewardship (AMS) in agriculture, particularly in food producing animals, to preserve the efficacy of these medicines.

The Food and Agricultural Organization of the United Nations (FAO), the World Organization for Animal Health (OIE), and the World Health Organization (WHO) developed a global action plan on AMR and urged member countries to make national action plans to address AMR [5]. As part of the Federal Action Plan, Canada has strengthened the regulatory framework on veterinary medicines to reduce AMU in food and agriculture. As of December 2018, all medically important antimicrobials for veterinary use are sold by prescription only [6]. However, dairy farmers with a valid veterinarian–client relationship are allowed to initiate antimicrobial treatment without veterinary consultation on the specific case. In the province of Quebec, the use of category I antimicrobials (i.e., 3rd and 4th generation cephalosporins, fluoroquinolones, and polymyxin) in food-producing animals, has been restricted to clinical cases that are not treatable with an antimicrobial of a lower importance category [7] and may only be used if prescribed by a veterinarian after attending to the individual animal in question. Similar restrictions might eventually follow in the rest of Canada.

Currently, more than 10,000 dairy farms in Canada with approximately 980,000 cows produce ~9 billion liters of milk annually [8]. The average herd size is 93 milking cows per farm, the proportion of farms with free-stall barn design is 35%, and the proportion of farms with automated milking systems is 12% [8]. Less than 2% of the milk is produced on certified organic farms [8]. Consequently, most dairy farms use antimicrobials regularly. A recent qualitative study of Canadian dairy farmers found that personal (also called internal) factors such as knowledge, experience, and satisfaction were the main drivers for AMU and that external factors such as veterinary advice had a smaller role in AMU decision-making [9]. This means that understanding farmers’ internal factors that influence AMU behavior is necessary to improve antimicrobial stewardship on dairy farms.

Even though economic factors are considered in AMU decisions, farmers are highly influenced by risk perception, values, and emotions [10]. Risk perception refers to people’s subjective judgments about the likelihood of negative outcomes [11]. It is crucial in health interventions because it is a strong determinant of people’s behavior. Previous studies indicate that farmers are less willing to modify their management practices if they do not see or accept the negative implications of their current behavior [12].

Values are conceptions of desirable end states. They are socially driven and are grounded in social and biological needs [13]. Values transcend specific actions and circumstances and serve as criteria to judge situations to decide what is good or bad, shape the interpretation of information, and influence attitudes and behaviors [13,14]. According to Schwartz’s model, there are ten values that can be described according to the motivational goal that underlies each of them: (1) self-direction; (2) stimulation; (3) hedonism; (4) achievement; (5) power; (6) security; (7) conformity; (8) tradition; (9) benevolence; and 10) universalism. The ten values can be reduced to four dimensions, grouping values that are motivationally close. Dimensions include conservation (security, conformity, tradition), self-transcendence (benevolence, universalism), openness to change (self-direction, stimulation, hedonism), and self-enhancement (achievement, power) [14]. People prioritize different values as more or less important to them and will likely behave in accordance with their most relevant values [15]. For that reason, it is necessary to align interventions for behavior change with values. Values have been studied in the context of AMS in human medicine [16,17], offering important insights for AMS in food animals. The current approach to AMS does not account for societal roles, cultural values, informal knowledge, and other social and cultural concerns that shape people’s actions [17]. Therefore, a values-focused reorientation of AMS could offer better results in reducing AMU [15]. There are data on dairy farmers’ attitudes toward AMU and AMR in Canada [9,18] and other countries [19,20]. However, to our knowledge, the association between individual values and AMU practices and attitudes toward AMU and AMR has not been investigated. Considering the need to understand internal drivers for effective implementation of AMS, our objectives were to describe dairy farmers’ decision-making process for AMU and their AMR risk perception, and to identify associations of values with attitudes toward promotion of prudent AMU among Canadian dairy farmers.

## 2. Materials and Methods

### 2.1. Questionnaire Design

We designed a questionnaire based on the results of a previous qualitative study [9], the literature available, and the experience of the researchers. The questionnaire was tested by five dairy farmers for comprehension, then refined based on their feedback. The complete survey consisted of 151 questions related to dairy cattle health management. Here we present results related to (a) general information on the respondents (6 items), (b) farm characteristics (7 items), (c) antimicrobial use decision-making process (9 items), (d) veterinarian–farmer relationship (6 items), (e) AMR risk perception and attitudes toward AMU reduction (15 items, with 5-point Likert scales from strongly agree to strongly disagree), and (f) individual values (20 items). The order of each section was randomized for each respondent. Questions included in this analysis are available in Appendix A. Items not included (e.g., sections on management of cull cows or non-ambulatory cattle) do not fit the scope of the present study.

Individual values (universalism, self-direction, stimulation, hedonism, achievement, power, security, conformity, tradition, and benevolence) were assessed using a validated instrument (the Twenty Item Values Inventory) containing two items per value [21]. Each item consisted of a sentence describing a person that implicitly represented a specific value orientation [14,21]. For example, item 1 stated: “S/he believes s/he should always show respect to his/her parents and to older people. It is important to him/her to be obedient”, representing the value of conformity. Participants were asked to respond on a 6-point scale, ranging from 1 (not like me at all) to 6 (very much like me), to indicate how similar they felt themselves to be.

### 2.2. Survey Distribution

This cross-sectional survey was distributed among dairy farmers in Ontario and Atlantic Canada (New Brunswick, Nova Scotia, and Prince Edward Island) between December 2020 and March 2021 using the Qualtrics XM software (Qualtrics, Provo, UT, USA). Participants were recruited by email through the provincial milk marketing boards to which producers belong. To comply with privacy regulations, we did not have access to the mailing lists or contact information of farmers. The intent was that all farmers with an email address would receive a message inviting them to participate in the survey. Reminders were distributed on printed cards included with a farmers’ magazine and through social media (Twitter account of SL and Dairy at Guelph, a university research network). Because we cannot quantify how many of the approximately 3500 farms in the target population were reached, it is not possible to specify a response rate. As this was an exploratory study, we did not calculate a minimum sample size. The first 250 respondents could participate in a draw to win one of two tablet computers. Identifying contact information for incentives was kept separate from all survey responses.

### 2.3. Data Analysis

Descriptive statistics and frequency tables were generated to explore the distribution of data. Associations between variables were explored using Pearson χ^2^ or Fisher’s exact tests for categorical variables. *t*-tests and Wilcoxon rank–sum tests were used to explore mean differences between categories when the continuous variable was or was not normally distributed, respectively. Answers on a 5-point Likert scale were classified as agree (for strongly or somewhat agree), disagree (for strongly or somewhat disagree), or neither (as is).

Values scores were analyzed using mean-centered scores (i.e., each individual’s responses were centered around his or her mean response to all 20 items) to control for individual differences [21]. The average of mean-centered scores for individual values and dimensions were calculated.

A factor analysis was performed on the 15 statements related to AMR awareness and AMU reduction in order to reduce the number of variables to several uncorrelated latent factors or dimensions. Considering the ordinal nature of our data, factor analysis was performed using a polychoric correlation matrix instead of raw variables. Visual (scree plot) and statistical (Horn’s parallel test) criteria were used to select the number of factors retained in the analysis.

A logistic regression model was used to identify variables associated with the probability of disagreeing with the need to promote responsible AMU in the dairy industry (the dependent variable was the yes or no answer to one question: Should there be more initiatives to promote responsible use of antibiotics in the dairy industry?). Unconditional associations with *p* < 0.10 were included in a multivariable model. Backward stepwise selection of the variables was made with *p* < 0.05 as the cut-off to be retained in the final model. Nominal variables gender, age category, and region (Ontario or Atlantic) were explored as potential confounders, but none of them met the specified criteria for confounder [22]. First-order interactions between all significant variables were explored, and no significant interactions were found. The Hosmer–Lemeshow goodness of fit test showed no indication of lack of fit of the model (*p* = 0.25). The coefficients, standard errors and confidence intervals of logistic regression were exponentiated to express them as odds ratios. We did not impute or adjust for missing responses. All statistical analyses were conducted using the software Stata 17 (StataCorp, College Station, TX, USA).

## 3. Results

### 3.1. Demographics and Farm Characteristics

Of 315 people who opened the survey link, 299 completed at least 50% of the survey. In total, 193 respondents were involved in AMU decisions on their farms and their data were used here. Most respondents identified themselves with the pronoun he/him (69%), were in the age group from 30 to 49 years old (53%) and had a college or university degree (69%). The most frequent role of respondents was owner or co-owner (76%). Respondents’ farms were mostly conventional (non-organic, 97%), with free-stall barn (69%) and parlor milking systems (44%). Most of the farms (65%) had more than one scheduled visit with a veterinarian per month. The average (SD) herd size was 115 (±103) milking cows, and the mean (SD) milk production per cow was 35 (±8) liters/day. The detailed demographics and farm characteristics of respondents are presented in Table 1. There were no significant differences between respondents in Ontario or Atlantic Canada in the demographic or farm characteristics studied (*p* > 0.05).

### 3.2. AMU Decision-Making Process and Veterinarian–Farmer Relationship

Most farmers (91%) indicated having Standard Operating Procedures (SOP) for AMU for common diseases. However, 23% of farmers did not follow the SOP in at least three of their last ten antimicrobial treatments. Animal body temperature was selected as the most important factor considered when deciding whether to treat an animal with antimicrobials by 48% of the respondents, followed by the attitude of the animal (22%), response to non-antimicrobial treatment (17%), time since signs or symptoms started (5%), previous antimicrobial treatment in the same animal (4%), and other factors (2%).

Farmers’ own experience and previous recommendations by their veterinarian were ranked as the most important considerations when choosing a particular antimicrobial product to use in calves and cows (Figure 1). The three most important reasons participants considered when selecting a particular antimicrobial product for calves and cows are shown in Figure 1. None of the antimicrobial use decision-making process variables were associated with demographic or farm characteristics (*p* > 0.05).

In addition, 99% of respondents indicated not always contacting their veterinarian before using antimicrobials in their cattle, and 11% of them never consulted their veterinarian before using antimicrobials on their farms. The most common reasons for not contacting the veterinarian were that they contacted their veterinarian as frequently as they wanted to (77% of respondents), followed by the ease or speed of communication (11%), and the cost of veterinary services (8%), among other reasons (4%). Most farmers (90%) affirmed that their veterinarian explained how to use antimicrobial treatments when the veterinarian is not consulted, and 32% of respondents agreed or strongly agreed that their veterinarian encourages them to use antimicrobials on their farm. Furthermore, 96% of participants considered that their veterinarian offers recommendations tailored to their farm’s needs, and 90% indicated they were satisfied with their veterinarian.

### 3.3. Farmers’ Attitudes toward AMR and AMU Reduction

Respondents to this block of questions (n = 151) were younger than the non-respondents of this block of questions (*p* = 0.008). To the question “Should there be more initiatives to promote responsible use of antibiotics in the dairy industry?” 23% of respondents answered no and 77% responded yes. Farm owners were more likely than employees (37% vs. 8%; *p* = 0.01) to consider additional promotion of AMS unnecessary, as well as respondents with tie-stall barns compared to responders with free-stall barns (42% vs. 17%; *p* < 0.001). Considering the additional promotion of AMS unnecessary was also more common among respondents who selected their own experience as the most important factor when deciding about antimicrobial treatments compared to respondents who selected others’ advice in the same question (36% vs. 18%; *p* = 0.02).

Although more than 70% of farmers agreed that they could explain what AMR is to a neighbor, 32% of them disagreed that AMU on their farm could cause AMR on their farm, almost half (47%) disagreed that AMU on their farm could cause AMR on other farms, and 38% disagreed that AMU on their farm could cause AMR in humans (Figure 2). The summary of the participants’ level of agreement with the 15 statements regarding AMR awareness, AMR knowledge, and AMU reduction is presented in Figure 2.

Providing more education to change or reduce AMU on dairy farms was selected by 45% of respondents as the best approach to promote responsible AMU, followed by measuring use to compare AMU among dairy farms (29%) and providing incentives to reduce AMU (23%). Only 2% of participants considered that new regulations would be a good strategy to reduce AMU on dairy farms. In addition, 48% of the participants considered that veterinarians should take the lead on promoting responsible AMU in the dairy industry, followed by milk boards (35%), individual farms (8%), pharmaceutical or animal health companies (7%), or others (5%).

The factor analysis conducted with the 15 variables presented in Figure 2 resulted in four factors that explained 60% of the variance of the respondents’ level of agreement with the statements. The rotated factor loadings for the four dimensions are presented in Table 2. Based on the value loading of the variables on each factor, four latent dimensions could be described as follows:Factor 1: knowledge and awareness of AMU implications for AMRFactor 2: sense of responsibility when using antimicrobials on their farmsFactor 3: inaccurate knowledge of AMR and low awareness of the AMR problem on dairy farmsFactor 4: fear of negative consequences of AMU reduction on cattle health and welfare

Scores for factor 1 were positively associated with considering that their veterinarian encourages AMU on farms (*p* = 0.04). Scores for factor 2 were positively associated with considering it unnecessary to increase promotion of AMS (*p* = 0.02). Scores for factor 3 were positively associated with being an employee of the farm (*p* = 0.05).

### 3.4. Individual Values

There were 175 responses to the values section of the survey. The demographics of these respondents did not differ significantly with the overall demographics described in Table 1. The value with the highest mean-centered score was benevolence, followed by self-direction and security. Tradition, stimulation, and power had the lowest mean-centered scores (Table 3). Self-transcendence was the overarching dimension with the highest mean-centered score, and self-enhancement had the lowest (Table 3). The mean-centered scores for benevolence were higher among participants using the pronoun she/her (*p* = 0.01). Being an employee of the farm (vs. being an owner) was associated with higher mean-centered scores for hedonism and security (*p* = 0.04 and *p* = 0.05, respectively). Participants younger than 29 years old scored higher for stimulation (*p* = 0.01).

Disagreeing with the need to promote responsible AMU in the dairy industry was associated with higher mean-centered scores for conformity (*p* = 0.01) and tradition (*p* = 0.02), and therefore with higher mean-centered scores for the dimension conservation (*p* = 0.01). Greater mean-centered scores for the dimension openness to change tended to be associated with agreeing to promote reduction of AMU in dairy industry (*p* = 0.06). In addition, greater mean-centered score for conformity was associated with higher score for factor 2 (*p* = 0.02), the mean-centered score for universalism was positively associated with score for factor 4 (*p* = 0.01), and the stimulation mean-centered score was positively associated with the mean-centered score for factor 1.

### 3.5. Logistic Regression Model

The final model for participants’ disagreement with greater promotion of responsible AMU on dairy farms was calculated from 129 observations. Respondents’ demographics included in the final model did not differ from the overall demographics presented in Table 1. The coefficients of the final logistic regression model are presented in Table 4. The model indicated that farm owners were more likely to consider it unnecessary to increase the promotion of responsible AMU in the dairy industry (OR: 6.7; 95% CI: 1.5 to 30.8), as did farmers with tie-stall barns (OR: 6.4; 95% CI: 2.0 to 21.1) and those who considered their own experience as the most important factor in selecting antimicrobial treatments (OR: 6.0; 95% CI: 1.8 to 20.7). A one-unit increase in the conservation value mean score (for example, from score 4 “somewhat like me” to score 5 “like me”) resulted in 1.7 times greater odds (CI: 1.2 to 2.3) of disagreeing with the need to increase promotion of responsible AMU in the dairy industry (Table 4).

## 4. Discussion

We examined dairy farmers’ decision-making on AMU, their perception of AMR risk, their values, and the association of these factors with their attitudes toward promoting responsible AMU in the dairy industry. Our results could inform development of interventions for AMS in the dairy industry. Specifically, our results indicate that a generic strategy to reduce AMU on Canadian dairy farms might not be sufficient due to individual differences in attitudes, values, and AMR awareness that shape AMR risk perception and willingness to modify current AMU practices.

Risk perception is affected by the cognitive dimension (i.e., how much farmers know and understand the risk of AMR) and the emotional dimension (i.e., how people feel about AMR) [11]. Similarly, we identified four factors that distinguish specific cognitive and emotional support needs to improve AMS in the dairy industry. For example, inaccurate knowledge of AMR based on responses to three questions that formed a cluster identified as factor 3 in Table 2 (e.g., incorrectly considering that antibiotics without milk withdrawal time are less likely to cause AMR) was associated with considering it unnecessary to increase promotion of AMS. Therefore, farmers with high scores for factor 3 would need information to change their knowledge about development of AMR. On the other hand, factor 4 reflects farmers’ fear of the negative implications of reducing AMU for animal welfare and health. Farmers with a high score in factor 4 would need a different approach with information on health management strategies to implement prudent AMU without compromising animal health, productivity, or welfare. Interestingly, although factor 2 was related to a high sense of responsibility when using AMU, it was associated with considering greater promotion of responsible AMU unnecessary (Table 4). Farmers with high scores in factor 2 might consider that all farmers have the same level of knowledge and sense of responsibility, leading them to consider it unnecessary to spend additional time, energy, and resources in promoting responsible AMU. These findings point to a need to tailor messaging and knowledge transfer to dairy farmers’ baseline knowledge and attitudes. We did not measure the AMU on respondents’ farms, and additional studies associating the factors that we evaluated with AMU would be desirable. In addition, it could be useful to classify dairy farmers with a few items in a questionnaire and validate tailored interventions. We underline that knowledge transfer alone (even if tailored) is unlikely to effect change in AMU behaviors, because as discussed below, there are other important influences on farmers’ AMU decision making.

Farmers with free-stall barns were more likely to agree with the greater promotion of responsible AMU than farmers with other types of barns. Generally, free-stall housing is associated with larger herd size and more personnel including family members and employees. Consistent with national statistics [8], the average number of milking cows on farms with free-stall barns was larger than tie-stall barns (132 animals versus 49 animals, respectively). We speculate that having more people involved in detection and potentially treatment of sick animals affects the perceptions of the need for structures for AMU. Additionally, this association with free-stall housing could be related to the fact that farmers consider that investing in technology and new barn design is important to reducing AMU, as previously reported in [22]. Tie-stall barns are reported to have higher incidence of Streptococcal and Staphylococcal mastitis [23,24,25] which is commonly treated with intra-mammary antibiotics. This might increase the perceived need for AMU and perhaps consequently more resistance to greater promotion of responsible AMU. However, the association with free-stall housing may relate to factors not directly tied to the housing system, such as the age of the facility, management practices, or milking process, among others [26]. Prudent AMU does not require switching from tie-stall to free-stall barn design or large investments in advanced technology. It may be necessary to help farmers to understand that judicious AMU can be achieved by improving biosecurity, practical selection of cases to treat with antibiotics, supportive rather than antimicrobial therapy in some cases, and rational drug selection and use [27].

Being a farm owner was associated with disagreeing with the need to increase promotion of responsible AMU in the dairy industry. Farm owners may be more concerned than employees about the possible economic impact of reduced AMU, additional workload, and the mechanisms for promoting responsible AMU. Essentially all respondents disagreed with implementing new regulations related to AMU. That is consistent with findings from our recent focus group study of Canadian farmers [9]. Dairy farmers considered their own experience as the main factor to decide about AMU in cows and calves (Figure 2). Therefore, we infer that they value their freedom to start individual antimicrobial treatment without consulting the veterinarian, which we further discuss below. Thus, farm owners might be opposed to further promoting responsible AMU due to concerns about new regulations that prevent them from deciding on antimicrobial treatments by themselves. Additionally, farm owners could be more influenced by the “good farming” concept. Good farming is related to cultural capital that is seen as prestigious by their social group, which influences their behavior [28]. Treating their animals by themselves is a way of demonstrating their skills, knowledge, and experience, which are symbols of good farming.

To our knowledge, this is the first study formally assessing individual values among dairy farmers in Canada. We employed the Twenty Item Values Inventory, which is used in the bi-annual European Social Survey [29] and has been used to characterize values in Swiss farmers [30]. We identified that benevolence had the highest mean score in our sample. This value is related to the preservation and enhancement of the welfare of their close ones and their social circle [14]. Taking into account that values shape the processing of information received [13], motivation strategies to achieve prudent use of antimicrobials in farmers with high benevolence scores should focus on the risk of AMR for them, their family, and their social circle instead of AMR as a global problem. A similar approach has been proposed for climate change, in which the problem should be rephrased as either a security issue for people with strong individualistic values (e.g., hedonism), or as an issue causing pain and suffering for humans and animals for people with strong communitarian values (e.g., benevolence and universalism) [13]. AMR and climate change have in common that their impacts are perceived as distant and that will affect others rather than an immediate problem that could affect oneself, which compromises the efficacy of AMS programs [15]. This dilemma emphasizes the need to re-structure current messages about AMR to farmers and the general public.

Individual values mean-centered scores differed between genders, age categories, and roles on the farm. This emphasizes the need to target interventions and to design specific strategies for different groups of farmers. For example, the youngest participants had higher stimulation scores. Thus, motivation strategies should include the novelty and challenging aspect of modifying current practices. On the other hand, a higher score for the conservation dimension was associated with disagreeing with the greater promotion of responsible AMU in the dairy industry. This is expected because conservation is related to preserving existing social arrangements that give certainty to life, thus this value conflicts with finding new solutions to existing problems [14].

Individuals with high scores for the dimension conservation might be motivated through benchmarking strategies (i.e., comparisons of the quantity of AMU between farms). Farm-specific benchmarking of AMU might motivate more conservative farmers to reduce AMU in pursuit of social desirability status [13]. One third of respondents agreed that benchmarking would be a good strategy to reduce AMU, indicating its acceptance among dairy farmers. Appealing to intrinsic values to motivate prudent AMU is more likely to succeed than forced behavioral changes [31].

Veterinarians were the most important external factor for AMU decisions, as reported in previous studies [18,32]. In addition, most participants considered that veterinarians should be responsible for promoting responsible AMU on dairy farms. Despite a vast majority of participants being satisfied with their veterinarian, essentially none always contacted their veterinarian before treating their cattle with antimicrobials, and 11% never contacted their veterinarian when deciding to use antimicrobials on their farms. Most participants indicated contacting their veterinarian as much as they wanted, indicating a lack of desire or perceived need to contact the veterinarian more often. The use of psycho-social models such as the Theory of Planned Behavior indicate that farmers are more likely to be influenced by “in-group” members rather than “out-group” members [28]. Veterinarians could be perceived as out-group members who do not share the experience or perspective of day-to-day farming, which may affect farmer–veterinarian communication. This is supported by our previous qualitative research, in which some farmers expressed frustration trying to understand messages delivered by veterinarians [9].

Similarly, difficulty in communication was the main barrier to contacting the veterinarian among farmers who would like to consult more frequently. We speculate that this implies lack of prompt direct access to veterinarian, perhaps by phone or text, at the time the farmer detects a non-severely ill animal. Ideally, veterinarians would be involved in diagnosis and treatment decisions of every case of illness, even if not directly or in-person. However, this approach is not always feasible due to costs, ease of communication, and veterinarians’ availability. Instead, a collaborative approach to AMS, in which veterinarians and farmers design farm-specific protocols for treating animal disease and periodically review all antimicrobial treatments and provide in-person feedback could be successful to achieve prudent AMU. Although some protocols for treatment are required for Canadian dairy farmers and possessed by many respondents, our results show that they are often not directly used in AMU decisions. Commonly, these protocols are developed by the veterinarian without direct input from the farmer. Our results suggest that a more participatory approach would likely appeal to the farmers’ empirical decision making and their self-direction, and so potentially achieve a greater buy-in and use of this standardized approach. This collaborative approach of feedback is less intrusive than restricting farmers’ freedom to treat their animals and therefore is more likely to succeed. The collaborative approach has been demonstrated to modify antimicrobial prescription behavior in veterinarians [27] and could be useful to improve AMU on Canadian dairy farms.

We acknowledge some limitations of this research. It is important to note that this survey was distributed during the COVID-19 pandemic, which may have affected the response rate. We acknowledge that the length of the full survey may also have been a deterrent to participation, although a survey of similar length of the same population achieved a satisfactory response [33]. As with any voluntary survey with impersonal contact for recruitment, this study could be subject to self-selection bias. It is possible that farmers with more information or more interest in the topic were more willing to participate. Farmers without post-secondary education and with tie-stall barns were under-represented in our study relative to the target population and that could potentially influence the results. Thirty percent of respondents were female which is an under-representation of the general population but is consistent with the proportion of female farm operators in Canada [34]. We did not employ methods to prevent multiple participation. However, there were no repeated IP addresses among participants or duplicated herd composition in our data. This is an exploratory study and additional research on how farmers’ values and attitudes can inform development and implementation of programs for achievement of AMS on dairy farms is desirable.

## 5. Conclusions

In conclusion, internal factors such as knowledge, experience, and values influence Canadian dairy farmers’ decisions regarding AMU. Therefore, promoting prudent AMU on dairy farms would likely benefit from targeted strategies that engage different motivations, values, and knowledge levels of dairy farmers.

## Figures and Tables

**Figure 1 antibiotics-11-00550-f001:**
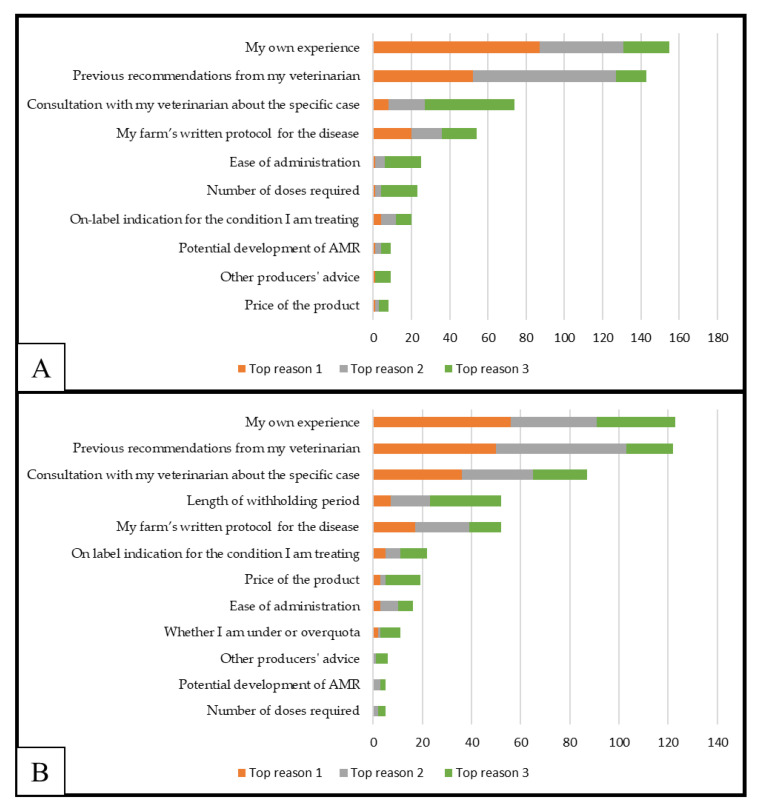
Three most important reasons ranked by 193 dairy farmers when selecting a particular antimicrobial product to treat calves (**A**) or lactating cows (**B**).

**Figure 2 antibiotics-11-00550-f002:**
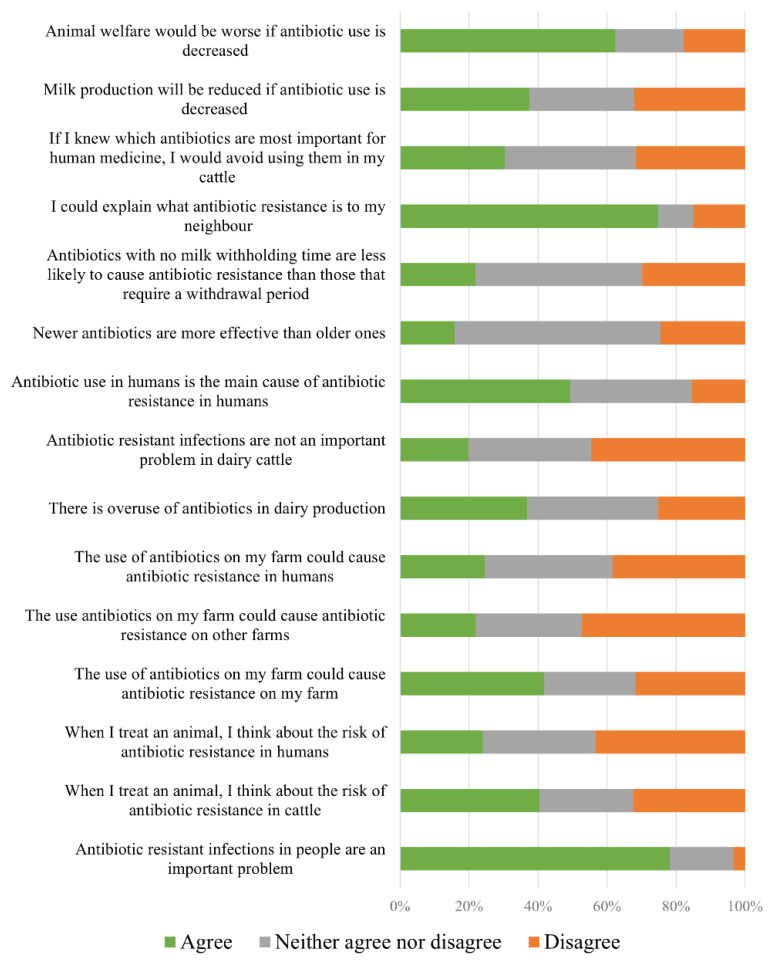
Summary of the 151 dairy farmer respondents’ level of agreement to 15 statements regarding antimicrobial resistance and antimicrobial use reduction.

**Table 1 antibiotics-11-00550-t001:** Demographics and farm characteristics of 193 respondents to a questionnaire on antimicrobial use by dairy farmers in eastern Canada.

	Respondent’s Region		
Demographics variables and farm characteristics	Ontario n = 156	Atlantic n = 37	Overall n = 193
	n (%)	n (%)	
Pronoun used			
He/Him	103 (71)	18 (60)	121 (69)
She/her	41 (28)	11 (37)	52 (30)
They/them	1 (1)	1 (3)	2 (1)
Age			
<29 years	30 (19)	2 (5)	32 (17)
30–39 years	48 (31)	13 (35)	61 (32)
40–49 years	33 (21)	8 (22)	41 (21)
50–59 years	33 (21)	6 (16)	39 (20)
>60 years	12 (8)	8 (22)	20 (10)
Highest level of education completed			
Primary school	12 (8)	4 (11)	16 (8)
High school	23 (15)	6 (16)	29 (15)
Post-secondary	112 (72)	22 (59)	134 (69)
Graduate education	9 (6)	5 (13)	14 (7)
Role on the farm			
Owner or co-owner	119 (76)	27 (73)	146 (76)
Employee	37 (24)	10 (27)	47 (24)
Farm characteristics			
Barn type			
Free-stall	105 (68)	28 (76)	133 (69)
Tie-stall	42 (27)	7 (19)	49 (26)
Bedded pack	8 (5)	2 (5)	10 (5)
Milk system			
Parlor	69 (45)	15 (41)	84 (44)
Pipeline	43 (28)	7 (19)	50 (26)
Automated milking system	43 (28)	14 (38)	57 (30)
Organic farm			
Yes	5 (3)	1 (3)	6 (3)
No	150 (97)	36 (97)	186 (97)
Frequency of scheduled visits of herd veterinarian			
The farm does not have scheduled visits with a veterinarian	12 (8)	2 (5)	4 (2)
Less than once a month	6 (4)	3 (8)	15 (8)
Once a month	36 (23)	8 (22)	44 (23)
More than once a month	102 (65)	24 (65)	126 (65)

**Table 2 antibiotics-11-00550-t002:** Varimax rotated factor loadings > 0.40 from principal components factor analysis of 5-point Likert scale responses by 151 dairy farmers to 15 statements regarding antimicrobial use and resistance. The analysis identified 4 latent dimensions or clusters in the data.

Dimension	Variables (Degree of Agreement with the Statement in a Questionnaire)	Rotated Factor Loadings
Factor 1 Knowledge and awareness of AMU implications for AMR	The use of antibiotics on my farm could cause antibiotic resistance on my farm	0.82
	The use antibiotics on my farm could cause antibiotic resistance on other farms	0.76
	The use of antibiotics on my farm could cause antibiotic resistance in humans	0.72
Factor 2 Sense of responsibility when using antimicrobials on their farms	When I treat an animal, I think about the risk of antibiotic resistance in cattle	0.70
	When I treat an animal, I think about the risk of antibiotic resistance in humans	0.69
	Antibiotic resistant infections in people are an important problem	0.81
Factor 3 Inaccurate knowledge of AMR and low awareness of the AMR problem on dairy farms	I could explain what antibiotic resistance is to my neighbor	0.81
	Antibiotic resistant infections are NOT an important problem in dairy cattle	0.49
	Antibiotics with no milk withholding time are less likely to cause antibiotic resistance than those that require a withdrawal period	0.71
	Newer antibiotics are more effective than older ones	0.50
Factor 4 Fear of negative consequences of AMU reduction on cattle health and welfare	Milk production will be reduced if antibiotic use is decreased	0.74
	Animal welfare would be worse if antibiotic use is decreased	0.80
	There is overuse of antibiotics in dairy production	−0.52
	We should reduce the use of antibiotics in dairy production	−0.56

**Table 3 antibiotics-11-00550-t003:** Mean-centered score and standard deviation (SD) for individual values and dimensions of the 175 dairy farmer respondents to the Twenty-Item Values Inventory evaluated in a 6-point Likert scale (0 = “not like me at all”; 6 = “very much like me”).

Value	Mean Centered Average Score (SD)	Dimension Mean Score (SD)
Benevolence	1.20 (0.20)	Self-transcendence
Universalism	1.03 (0.23)	1.11 (0.17)
Self-Direction	1.10 (0.23)	Openness to change
Stimulation	0.90 (0.24)	0.98 (0.16)
Hedonism	0.96 (0.22)	
Security	1.07 (0.18)	Conservation
Conformity	1.05 (0.25)	0.97 (0.17)
Tradition	0.80 (0.29)	
Achievement	0.96 (0.25)	Self-enhancement
Power	0.93 (0.25)	0.94 (0.19)

**Table 4 antibiotics-11-00550-t004:** Final logistic regression model for the odds of answering no to the question “Should there be more initiatives to promote responsible use of antibiotics in the dairy industry?” from 129 responses in a survey of Canadian dairy farmers.

Variables	Odds Ratio	95% Confidence Interval	*p*-Value
Role on the farm			
Employee	Referent		
Owner	6.7	1.5–30.8	0.015
Barn type			
Free-stall barn	Referent		
Tie-stall barn	6.4	2.0–21.1	0.002
Bedded pack	2.1	0.2–26.0	0.572
Participants’ most important factor in selecting an antimicrobial product			
Others’ advice	Referent		
Own experience	6.0	1.8–20.7	0.004
Score for factor 2 (sense of responsibility for AMU) from the PCA of responses to questions about AMU and AMR ^1^	1.8	1.1–3.3	0.040
Conservation value score ^2^	1.7	1.2–2.3	0.001

^1^ See text and Table 2 for details; PCA = Principal Components Analysis; odds ratio per 1 unit increase in average score for the 3 questions in this factor. ^2^ Mean-centered score from the Twenty Item Values Inventory; odds ratio per 1 unit increase in score for this value.

## Data Availability

Data that support the findings of this study are available from the corresponding author, upon reasonable request.

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
