# Peer review of "Values and Risk Perception Shape Canadian Dairy Farmers’ Attitudes toward Prudent Use of Antimicrobials"

_antibiotics, 2022, doi:10.3390/antibiotics11050550_

Round 1

Reviewer 1 Report

The study sought to determine dairy farmers' personal factors for AMU and risk perception and associate these factors with their attitudes towards promotion of prudent AMU. The findings will lead to formulation of strategies for promotion of AMU by dairy farmers in the study areas and possibly elsewhere.

Abstract:

line 14: Specify Atlantic provinces

line 31: Include AMR and AMU in key words

Line 47: Should read Food and Agriculture

Line 58: add to after attending

Lines 83-98: may not be necessary and could be summarized and referenced

Line 105: insert in between results and reducing

Fig.1: Consultation with my veterinarian about specific antimicrobial?

Results

Line 379: should read, as was previously reported rather than as we reported in (22)

Line 380: Tie-stall barns are reported to have higher incidence of ...

Line 384: should read age of facility instead of facility age

Line 399: the phrase should read, which we further discuss below...

Line 421: Delete full stop after programs

Line 428-430: sentence should be re-phrased to convey the point that is being made

Line 436: Redding et al. 2020 should be replaced with reference (13)

Discussion

Line 474: substitute the for our response rate

General: There is need to check out for appropriate punctuations in several places.

Author Response

Dear reviewer,

Thank you for your valuable comments and suggestions to improve our research paper. We addressed all your comments as indicated below in italics. 

Kind regards,

Claudia Cobo-Angel

Reviewer: The study sought to determine dairy farmers' personal factors for AMU and risk perception and associate these factors with their attitudes towards promotion of prudent AMU. The findings will lead to formulation of strategies for promotion of AMU by dairy farmers in the study areas and possibly elsewhere.

Abstract:

line 14: Specify Atlantic provinces

Authors: The provinces were specified (lines 12-13)

line 31: Include AMR and AMU in key words

Authors: The keywords were added. Thanks for your suggestion.

Line 47: Should read Food and Agriculture

Authors: The word food was added (line 44)

Line 58: add to after attending

Authors: The word “to” was added (line 52)

Lines 83-98: may not be necessary and could be summarized and referenced

Authors: We summarized and referenced the description of the values (Lines 76-81)

Line 105: insert in between results and reducing

Authors: The word “in” was added (line 92)

Fig.1: Consultation with my veterinarian about specific antimicrobial?

Authors: Figure 1 was replaced, and the complete sentence is now visible

Results

Line 379: should read, as was previously reported rather than as we reported in (22)

Authors: The sentence was modified (line 367)

Line 380: Tie-stall barns are reported to have higher incidence of ...

Authors: The sentence was modified (line 367)

Line 384: should read age of facility instead of facility age

Authors: The sentence was modified (line 372)

Line 399: the phrase should read, which we further discuss below...

Authors: The sentence was modified (line 386)

Line 421: Delete full stop after programs

Authors: The punctuation was corrected (line 408)

Line 428-430: sentence should be re-phrased to convey the point that is being made

Authors: The sentence was modified (line 416-419)

Line 436: Redding et al. 2020 should be replaced with reference (13)

Authors: The reference was corrected (line 425)

Discussion

Line 474: substitute the for our response rate

Authors: the sentence was modified (line 464)

General: There is need to check out for appropriate punctuations in several places.

Authors: punctuation was revised and adjusted.

Reviewer 2 Report

Cobo-Angel et al analyze the values and risk perception of Canadian dairy farmers's attitude towards judicious use of antimicrobials. Antimicrobial resistance is a growing concern globally and there are very few antimicrobials in production pipeline while AMR is growing. In North America, the antimicrobial usage among poultry farmers has been declining while Pig and Cattle farmers were not able to reduce the antimicrobial use that much. The authors have conducted a very important study to delineate the attitude of dairy farmers on AMU and it is evident that more strategies incorporating veterinarians is needed to curb AMR.

The manuscript is well presented with good statistical analyses. However, one question I have is there a chance for data duplication since multiple employees from a single farm could have responded from different IP address? How it could have affected the data? What can be done to prevent it?

Author Response

Dear reviewers,

Thank you for your valuable comments and suggestions to improve our research paper. 

Kind regards,

Claudia Cobo-Angel (on behalf of the authors)

Reviewer 2:

The manuscript is well presented with good statistical analyses. However, one question I have is there a chance for data duplication since multiple employees from a single farm could have responded from different IP address? How it could have affected the data? What can be done to prevent it?

Authors: The survey was distributed mainly by the provincial milk boards through their mailing list. It is unlikely that multiple farmers from the same farm are included in their mailing list. In addition, we did not see repetition of herd compositions. This was included in the manuscript (Line 474-476).

Reviewer 3 Report

Thank you for the opportunity, 
I found interesting and well written the manuscript, by C. Cobo - Angel et. all entitled "Values and risk perception shape Canadian dairy farmers' attitudes toward prudent use of antimicrobials".
Considering the AMR problem from a one healt perspective, the authors interviewed dairy farmers in Ontario and Canada to gain information on their attitudes towards antibiotic use. 
Although the data collected are subjective opinions, I believe this method is useful to better understand the actual use of antibiotics by farmers. 
Having found no criticalities in the manuscript I propose it for acceptance in this form.

Author Response

Dear reviewer,

Thank you very much for your review and acceptance suggestion.

Claudia Cobo-Angel (on behalf of the authors)